

# Two global climatologies of daily fire emission injection heights since 2003

S. Rémy[1], A. Veira[2], R. Paugam[3], M. Sofiev[4], J.W. Kaiser[5], F. Marenco[6], S.P. Burton[7], A. Benedetti[8], R.J. Engelen[9], R. Ferrare[10], and J.W. Hair[11]

[1]Laboratoire de Météorologie Dynamique, UPMC/CNRS, Paris, France
[2]Max Planck Institute for Meteorology, Hamburg, Germany
[3]King's College, London, United Kingdom
[4]Finnish Meteorological Institute, Helsinki, Finland
[5]Max Planck Institute for Chemistry, Mainz, Germany
[6]Observational Based Research, Met Office, Exeter, United Kingdom, U.K.
[7-10-11]National Aeronautics and Space Administration, Langley Research Center, Hampton, CA, U.S.A.
[8-9]European Centre for Medium-range Weather Forecasts, Reading, U.K.

*Correspondence to:* S. Rémy
(samuel.remy@lmd.jussieu.fr)

**Abstract.**

The Global Fire Assimilation System (GFAS) assimilates Fire Radiative Power (FRP) observations from satellite-based sensors to produce daily estimates of biomass burning emissions. It has been extended to include information about injection heights provided by two distinct algorithms, which also use meteorological information from the operational weather forecasts of ECMWF.

Injection heights are provided by the semi-empirical IS4FIRES parameterization and an analytical one-dimension Plume Rise Model (PRM). The two algorithms provide estimates for injection heights for each satellite pixel. Similarly to how FRP observations are processed in GFAS, these estimates are then gridded, averaged and assimilated, using a simple observation operator, so as to fill the observational gaps. A global database of daily biomass burning emissions and injection heights at 0.1°resolution has been produced for 2003-2015. The database is being extended in near-real-time with the operational GFAS service of the Copernicus Atmospheric Monitoring Service (CAMS).

The two injection height datasets were compared against a new dataset of satellite-based plume height observations. The IS4FIRES parameterization showed a better overall agreement against observations, while the PRM was better at capturing the variability of injection heights and at estimating the injection heights of large fires. The results from both also show a differentiation depending on the type of vegetation. A positive trend with time in median injection heights from the PRM was noted, less marked from the IS4FIRES parameterization. This is provoked by a negative trend in number of small fires, especially in regions such as South America.

The use of biomass burning emission heights from GFAS in atmospheric composition forecasts was assessed in two case studies: the South AMerican Biomass Burning Analysis (SAMBBA) campaign which took place in September 2012 in Brazil, and a series of large fire events in the Western U.S. in August 2013. For these case studies, forecasts of biomass burning aerosol



species by the Composition-Integrated Forecasting System (C-IFS) of CAMS were found to better reproduce the observed vertical distribution when using PRM injection heights from GFAS.

# 1 Introduction

## 1.1 Background and motivation

Vegetation fires are responsible for the release of massive quantities of trace gases and aerosols into the atmosphere (Andreae and Merlet (2001), Van der Werf et al. (2010)). Each year, around 3 million square kilometers are burnt worldwide (Giglio et al. (2010)) by landscape fires ignited by natural or anthropogenic causes. A wide range of atmospheric processes are affected by fire emissions, including radiative transfer, turbulent diffusion, cloud microphysics and atmospheric chemistry (Twomey (1977), Heald et al. (2014), Veira et al. (2015b) among others). Fires cause large and overall poorly quantified emissions
of particulate matter; they have been estimated by Bond et al. (2013) at 2-11 Tg per year (Black Carbon) and 18-77 Tg per year(organic carbon). Particulate matter emitted by fires can also occasionally cause acute air quality problems, such as in Singapore in June 2013 for example. Voulgarakis et al. (2015) found that biomass burning emissions are almost entirely responsible for the inter-annual variability of the global CO, OH radical and aerosol (not taking into account dust and sea-salt particles) abundances.

Several global fire emissions database have been developed in the recent years: FLAMBE, GFED, FINN, QFED, IS4FIRES and GFAS (Reid et al. (2009), Van der Werf et al. (2010) and Giglio et al. (2013), Wiedinmyer et al. (2011), Darmenov and da Silva (2013),Sofiev et al. (2009), Kaiser et al. (2012) respectively). All of these products rely on satellite observations, either of Fire Radiative Power (FRP), hot spots or burnt area, because they alone provide sufficient spatial coverage and temporal sampling frequency (Giglio et al. (2006), Kaiser et al. (2012)). Inverse modeling methods have also been used together with
satellite-based retrievals of CO (e.g. Gonzi et al. (2010),Krol et al. (2013) and Gonzi et al. (2015)) or Aerosol Optical Depth (AOD) (e.g. Ichoku and Kaufman (2005); Ichoku and Ellison (2014)) or a combination (Konovalov al., 2014) to estimate fire emissions. In this study, we will describe a recent addition to the Global Fire Assimilation System (GFAS) inventory.

  When a fire occurs, the intense heat from the burning vegetation creates a rising plume, which interacts with the ambient atmosphere and transports the fire emissions (Freitas et al. (2006), Paugam et al. (2015a)). Because of this fire-induced con-
vection, biomass burning is one of the very few natural processes that emit large quantities of aerosols and trace gases well above the Planetary Boundary Layer (PBL), sometimes even reaching the upper troposphere/lower stratosphere (Andreae et al. (2004), Fromm et al. (2005), Dahlkötter et al. (2013)). Using observations from the Multi-angle Imaging SpectroRadiometer (MISR) instrument onboard Terra, Kahn et al. (2008) estimated that 5 to 18% of fires that occurred over Alaska and Canadian Yukon in 2004 released smoke constituents above the PBL. Sofiev et al. (2012), using an injection height parameterization
found that 15% of fire plumes reached the free troposphere.

  Fire emission height, i.e. the height at which the fire smoke releases emissions into the atmosphere, has been identified as a crucial parameter to influence the forecasts of the life cycle of biomass burning aerosols. In particular, whether smoke constituents are released below or above the top of the PBL will have profound consequences on their transport, deposition



and life cycle more generally. It has been shown (Textor et al. (2006) that there are significant uncertainties in the vertical distribution of aerosols in global models, whereas this information is critical in estimating the magnitude of the direct radiative forcing (Choi and Chung (2014) for example).

As noted in Veira et al. (2015a), the expressions "fire emission heights", "injection height" and "plume height" have been used as equivalent terms in the literature although they can cover different realities. In this article, "injection height" describes the height at which most of fire emissions occur, i.e. the height of maximum injection.

The height at which smoke constituents are released depends on the intensity of the updraft generated by the fire, which itself depends on a variety of parameters (Kahn et al. (2007)); foremost among them are the sensible heat flux released by the fire, the size of the fire and the temperature and humidity profiles of the surrounding atmosphere. The ambient atmosphere has a twofold impact on the vertical transport in a fire plume: the thermal stratification acts more or less strongly against the buoyancy triggered by the sensible heat released by the fire. Ambient cooling also favors water vapour condensation, with the formation of pyro-cumulus and/or -cumulonimbus (Fromm et al. (2010)), and the release of latent heat which can accelerate the vertical transport in the smoke plume.

Several theories and models describing the dynamics of plume rise have been developed, with the objective to estimate either a single parameter (injection height) or the entire fire emission profile. The main algorithms available today are summed up in Paugam et al. (2015a) and Veira et al. (2015a). They can be divided into two families: semi-empirical models (Achtemeier et al. (2011) and Sofiev et al. (2012) for example) and analytical-numerical Plume Rise Models (PRM) such as the one developed by Latham (1994) and further refined by Freitas et al. (2007) and Paugam et al. (2015b). All of these models use as input atmospheric profiles of meteorological parameters together with satellite information on fire size and/or activity. Their output are either a injection height (for the Sofiev or IS4FIRES parameterization) or a full smoke detrainment profile (for the PRM). In this study, we will use the algorithm described in Sofiev et al. (2012), hereafter denoted as "IS4FIRES" and in Paugam et al. (2015b), hereafter denoted as "PRM".

## 1.2 Objectives

Our objective is to embed the two "IS4FIRES" and "PRM" algorithms into GFAS, so that the combined system provides daily estimates plume injection heights in addition to the already existing estimates of FRP, biomass burning emission of smoke constituents. Because there are still many uncertainties on how to estimate injection heights, two approaches have been chosen representing the most accepted methods currently. Each dataset was evaluated against observations, characterized for its advantages and limitations and is freely available online at http://apps.ecmwf.int/datasets/data/cams-gfas. The users of GFAS are now given a choice of which injection height to use, and also the possibility to combine the two.

The extended GFAS (hereafter simply called "GFAS" for clarity) was run for the whole period of time for which fire observations were available from the Moderate Resolution Imaging Spectroradiometer (MODIS) onboard the Aqua and Terra satellites, i.e. 2003 to the present date. A coherent climatology of daily biomass burning emissions and injection heights provided by two algorithms was produced. To our knowledge, this is the first time such a climatology was made available to





the public. The closest existing database is the monthly climatology of vertical profiles produced by Sofiev et al. (2013) using one of the algorithm used in this study.

The injection height database described in this paper can be useful for a number of applications, from climate studies to atmospheric composition modeling and to the detailed study of past fire events. Two examples of using daily injection heights

from GFAS in atmospheric composition modeling are presented in this paper. The resulting forecasts were compared against airborne observations of aerosol extinction.

A secondary objective was to compare the results of the two methods, which are representative of two very different approaches to estimating injection heights: semi-empirical and analytical. Veira et al. (2015a) already compared the output of IS4FIRES and the PRM against the MPHP dataset; here we aim to further this comparison. Valmartin (2012), Rosario et al.

(2013) and Strada et al. (2013) assessed various injection height algorithms in different situations and areas and found mixed results overall. This study will be an occasion to revisit their conclusions.

The structure of the paper reflects these objectives. A methodology section details the components of the system: GFAS, the IS4FIRES and PRM algorithms as well as the verification dataset from MPHP2. Another section presents how the injection heights algorithms were embedded in GFAS. Two sections describe the injection height database; one is dedicated to geo-

graphical and time analysis, as well as comparison against FRP and relevant meteorological parameters; another is dedicated to comparison against MPHP2 injection heights. A final section presents two applications of injection heights from GFAS using the C-IFS atmospheric composition model in two fire situations. The first one consists of the South AMerican Biomass Burning Analysis (SAMBBA) field campaign in Brazil in September 2012 (Angelo (2012)). The other concerns large fires in the US in August 2013, during the SEAC4RS field campaign. Forecasts of biomass burning aerosols were compared against

airborne observations for both situations.

## 2 Methodology

### 2.1 Input data and models used

#### 2.1.1 MISR plume heights: MPHP2

The MPHP plume height dataset is a combination of the MISR smoke aerosol and the MODIS MOD14 thermal anomaly

products. The latest release of the MPHP (April 2012) includes data of wildfire smoke plumes in North and South America, Eurasia, Africa and Southeast Asia, observed between 2001 and 2009. For more detailed information, see the official product description at http://www-misr.jpl.nasa.gov/getData/accessData/MisrMinxPlumes/. As stated in the MPHP data quality statement, important biases are introduced by pyro-cumulus clouds which hide below-cloud fire activity (Kahn et al. (2008)), by errors in the digitization of the plumes and by large uncertainties in the MODIS fire pixels.

The MPHP2 dataset that was released in July 2015 is using a new version of the satellite retrieval software, with the addition of a blue-band height retrieval. The MISR conventional red-band retrieval of height for optically thin smoke over relatively bright terrain is frequently unsuccessful and may under-estimate smoke height. Height retrievals for many plumes in the





MPHP2 dataset used the blue-band retrieval, resulting in a significant increase in the number of successful plumes. The MPHP2 dataset includes over 32000 fire plumes for the year 2008. When excluding plumes of poor retrieval quality, the MPHP2 dataset provides 13454 injection heights which we will use to validate the injection heights from the two algorithms embedded into GFAS.

According to the official MPHP product description as well as Kahn et al. (2008) and Nelson et al. (2013), an observational plume height accuracy of about 200 m can be assumed in good conditions. In the absence of a different prescription for the new dataset, we will use the same value for the MPHP2 dataset.

### 2.1.2 Plume Rise Model (PRM)

The PRM comes in three versions: v0, described in detail in Freitas et al. (2007), v1 and v2 which are introduced in Paugam

et al. (2015b). In this work, the latest version of the PRM is used and the term "PRM" will denote this particular version. The Plume Rise Model consists of a 1-D cloud-resolving model, forced at its base by satellite-derived fire parameters: Convective Heat Flux (CHF), and Active Fire area (AF-area). To account for possible condensation and formation of pyro-cumulus, the PRM also includes a bulk microphysical scheme based on Kessler (1969). The latest version of the PRM also includes a parameterization of horizontal mass exchange. Four prognostic variables: vertical velocity; temperature; horizontal plume

velocity and the radius of the plume are forecasted by the model. These 4 variables are governed by equations based on the vertical motion and mass conservation equations as well as the first thermodynamic law (Freitas et al. (2007)).

The PRM is run using a 100 m resolution vertical grid, reaching a maximum height of 20km. The time step is adaptive, computed to respect the Courant-Friedrich-Levy numerical stability criterion, with an upper limit of 5 seconds (Freitas et al. (2007)). The ambient atmospheric profiles are taken from the European Centre for Medium Range Forecasts (ECMWF) data.

Besides the atmospheric profiles, the main inputs of the PRM are Convective Heat Flux (CHF) and Active Fire (AF)-area. CHF is defined as :

$$CHF = \beta FRP \tag{1}$$

Where $\beta$ is a scaling factor that is fitted. FRP can be either provided directly by the MODIS MOD14 product (Justice et al. (2002) and Giglio (2005)), or derived from mid-infrared emissivity observations from MODIS as detailed in Paugam et

al. (2015b). The fire temperature and area (the AF-area parameter) are estimated from MODIS observations in the mid- and thermal- infrared channels using the Dozier algorithm (Dozier (1981)) on clusters of contiguous active fire pixels. Injection heights are computed only for fires with an area above 1 ha, and a MODIS sub-pixel effective temperature above 600K, to prevent taking into account smoldering fires behind the main fire front.

### 2.1.3 IS4FIRES

The semi-empirical IS4FIRES parameterization is detailed in Sofiev et al. (2012). Injection height is estimated as a function of the PBL height, the Brunt-Väisälä frequency of the free troposphere and the total FRP of a fire. In our implementation, the PBL height is provided by the operational ECMWF model, which uses the Troen and Mahrt (1986) diagnostic: PBL height





is defined as the level where the bulk Richardson number, based on the difference between quantities at that level and the lowest model level, reaches the critical value 0.25. Evaluation of this product against satellite retrievals (Palm et al. (2005)) showed that the diagnostic often underestimates the PBL height by a few hundred meters but shows a good correlation with observations. In this work the two-step version of the IS4FIRES algorithm is used.

IS4FIRES was further refined by Kukkonen et al. (2014) and Veira et al. (2015a). Kukkonen et al. (2014) improved the results of the algorithm when replacing the Brunt-Väisälä frequency of the free troposphere by the inversion layer Brunt-Väisälä frequency in case of stable nocturnal boundary layer. Since our proposed implementation will use a daily FRP product that is based on day-time satellite observations from MODIS, this improvement was not tested as its impact would probably not be significant.

## 2.2 Integration in GFAS

### 2.2.1 Overview of the Global Fire Assimilation System (GFAS)

The European Union funded operational "Copernicus Atmospheric Monitoring Services" (CAMS) provides global analysis and forecasts of atmospheric composition, alongside European air quality forecasts (Hollingsworth et al., 2008) using the C-IFS model (Morcrette et al. (2009), Benedetti et al. (2009), Peuch and Engelen (2012) and Flemming et al. (2015)). In order to provide this forecasting system with accurate estimates of aerosol, reactive gases and greenhouse gas emissions from biomass burning, the Global Fire Assimilation System (GFAS, Kaiser et al. (2009, 2012)) uses satellite measurements of fire radiative power (FRP) to estimate daily fire emissions. GFAS is operated operationally withing the CAMS project and provides daily estimates of FRP, dry matter burnt and biomass burning emissions of 41 species. The data is freely available from http://apps.ecmwf.int/datasets/data/cams-gfas/ .

GFAS grids and averages FRP observations from the MODIS instrument onboard NASA's Terra and Aqua satellites. FRP observations from sensors onboard geostationary satellites such as Meteosat-8/-9/-10 and GOES East and West are not used in GFAS, as their values are very different from MODIS. This gridded data from the two satellites are then merged to produce global daily averaged FRP fields with 0.1 °resolutions. An analysis of daily averaged FRP is constructed by assimilating this merged daily averaged FRP observation. The assimilation step consists of a simple Kalman filter used with a persistence model. The objective of the data assimilation step is to correct for gaps in the observations, caused mainly by cloudy conditions (sometimes caused by pyro-cumulus or -cumulonimbus). Spurious FRP observations of volcanoes, gas flares and other industrial activity are masked.

Correlations between FRP and fuel consumption resp. fire emission were found in several contexts: Wooster et al. (2005) demonstrated this for fuel consumption in laboratory studies, Ichoku and Kaufman (2005) documented it for aerosol emission observed by satellite, and Heil et al. (2010) found strong, but biome-specific, correlations between FRP and the dry matter combustion rate of the Global Fire Emission Database (GFED, Van der Werf et al. (2010)) v3.1. The latter allowed the derivation of conversion factors for eight land cover classes that link GFAS FRP to GFED dry matter combustion rate, and subsequently the





estimation of global dry matter burnt fields with GFAS. Emission factors following Andreae and Merlet (2001) and updates are then used to estimate the emissions of 41 species from the dry matter burnt.

The GFAS emission inventory cover the period from 1 January 2003 to present. It has been recently extended to early 2000 (Remy and Kaiser (2014)) using bias-corrected observations from MODIS onboard Terra only. The output from GFAS is

validated regularly in the framework of the CAMS project (Andela et al. (2013)).

### 2.3   Integration of the injection height algorithms into GFAS

The output of the IS4FIRES parameterization is a single parameter: injection height, which makes it easy to assimilate once gridded and averaged. The output of the PRM is a whole detrainment profile, which would be too costly to assimilate. It was decided instead to derive the most useful parameters from this profile and then to regrid and assimilate them. These parameters

consist of the top and the bottom of the plume, and of the mean height of maximum injection, i.e. the average of the PRM levels for which detrainment is equal or above half of maximum detrainment.

The clustering algorithm of the PRM was used to produce clusters of contiguous MODIS fire pixels. The PRM is run on the accumulated FRP of each fire cluster, while IS4FIRES is using the maximum FRP of each cluster, so as to use inputs that are similar to Sofiev et al. (2012), who used maximum FRP from MPHP fire clusters in their work.

The output of the PRM is very dependent on the stability of the atmosphere. One possible drawback is that in some cases, the impact of the fire forcing at the base of the 1D column of the PRM becomes negligible as compared to the impact of the atmospheric environment. Then, the injection profiles produced by the PRM are only representative of the shallow convection scheme included in it, not of the fire input. To prevent this, the PRM is run twice, once with no fire forcing, and another time with the fire forcing. Only the fire clusters for which the mean height of maximum injection provided by the PRM forced by

the fire is larger than the one provided by the PRM not forced by the fire are kept. This particular criterion removes around 10% of active fire clusters.

The two injection height algorithms provide four parameters in all, for each 5mn MODIS granule: three for the PRM (mean height of maximum injection, height of the top of the plume and of the bottom of plume), and one for IS4FIRES. These parameters must first be gridded onto the 0.1 x 0.1 °GFAS grid. To achieve that, the number and coordinates of all the MODIS

pixels that constitute the fire clusters are kept. Each GFAS grid cell containing at least one pixel of a given fire cluster is then associated with the values of the 4 injection height of this fire cluster. In the unlikely case where a GFAS grid cell contains pixels from two or more distinct fire clusters, the maximum value from the fire clusters is assigned to the GFAS grid cell. Maximum FRP from each fire clusters is also gridded in the same way.

In GFAS (Kaiser et al. (2012)), the time averaging of the 5mn global gridded FRP into hourly and daily global gridded

FRP uses the fraction of satellite observed area as a weight. For the injection height parameters it was decided to use the gridded maximum FRP as a weight in order to privilege injection heights associated with the most active fires. This means that the resulting product is more representative of the diurnal maximum of fire intensity. The rationale behind this choice is that several studies showed that most of emissions from biomass-burning occur during or around the diurnal maximum (Andela et





al. (2015), Freeborn et al. (2009), Freeborn et al. (2011), Roberts et al. (2009)). The daily gridded fields of the four injection height parameters are then assimilated alongside FRP in the data assimilation step of GFAS.

The fact that there are several conditions for the PRM to produce a detrainment profile for a given fire cluster (selection based on fire temperature, on fire size and finally the check against the output from the non-forced PRM mentioned above) means that there are many GFAS grid cells with non-null FRP and biomass burning emissions with null injection heights. In all, only 33 to 35% of GFAS grid cells with positive fire emissions also have positive injection height parameters, as shown by Figure 2. Regions where this ratio is large, above 5, include China, India, parts of Ukraine and Russia, central US and South Brazil. They correspond to areas where the human presence is high; most of the fires occurring in these regions comes from agriculture. They are most of the times rather small in intensity, which explains why injection heights were not computed for these fires for over 80% of cases.

## 3 Results

### 3.1 Comparison of the raw output of the two methods

Figure 3 shows density plots of the injection height provided by the IS4FIRES parameterization against the boundary layer height diagnostic from the operational ECMWF model, for global Terra and Aqua observations and for the 1/6/2013 to 1/8/2013 period. The injection heights are neatly separated into two distinct subsets: one below the PBL height, with a large variability, and another just above PBL height with low variability. The two subsets are clearly a result of the two steps of the algorithm, with the second step being applied only to plumes that are rising higher than the PBL height. However, it is apparent from this plot that IS4FIRES encounters difficulties in providing injection heights that are significantly above PBL height.

The same plot also shows density plots of the mean height of maximum injection provided by the PRM against the boundary layer height diagnostic from the operational ECMWF model, for global Terra and Aqua observations and for the 1/6/2013 to 1/8/2013 period. The mean height of maximum injection is defined as the average of the PRM levels where plume detrainment is above half of maximum detrainment. The impact of the thermal stratification of the atmosphere is very clear: for most of the fire plumes, maximum detrainment occurs just under the top of the PBL, where a temperature inversion sometimes occurs and may block the vertical development of fire plumes. Maximum injection occurs higher with Aqua observations than with Terra. The Terra overpass time is around 10:30 local solar time in its descending mode and 22:30 local solar time in its ascending mode. The Aqua overpass times are around 13:30 (resp. 01:30) local solar time in ascending (resp. descending) mode.

The different values for maximum injection obtained with Aqua and Terra observations show that the two methods reproduce the maximum of the diurnal cycle of fire activity during the early afternoon well. With Terra observations, a number of cases with a very stable PBL give very low maximum injection heights; this is more marked for the PRM. With Aqua, the number of such cases is smaller.

Comparing the output of the two methods (lower panel of Figure 3) show that For Terra, values from the PRM are more often smaller than from IS4FIRES while for Aqua, the PRM gives more often always higher values. For both satellites, the correlation





between the results of the two algorithms is quite weak, and the dispersion is larger with the PRM than with IS4FIRES. This feature was already noted by Veira et al. (2015a) and Paugam et al. (2015b).

Figure 4 shows density plots of injection height from IS4FIRES, and of the mean height of maximum injection from the PRM, against observed FRP (from Aqua) and fire area as estimated by the Dozier algorithm. For both algorithms, the cor-
relation with the estimated fire area is very weak. The IS4FIRES injection heights shows a non-linear link with maximum FRP, until a plateau is reached for high maximum FRP values. Veira et al. (2015a) found that increasing the FRP input of the IS4FIRES parameterization has little impact on the estimated injection height, which is confirmed here for higher values. For the PRM, the statistical link between maximum FRP and injection height is close to null, which is coherent with the results of Paugam et al. (2015b). This also reproduces the very weak correlation between FRP and injection heights in the MPHP dataset
(Sofiev et al. (2012)).

## 3.2  Comparison of the two injection height climatologies

The extended version of GFAS that includes the injection height parameters has been run from 1 Jan 2003 to 1 Jan 2015. The resulting injection height climatologies are presented and evaluated in this section.

The average values of the mean height of maximum injection and of the plume top, from the PRM component of GFAS, and
of the injection height from the IS4FIRES component of GFAS are presented in Figure 5. For the sake of simplicity, these two components will be named as "PRM" and "IS4FIRES" respectively. Over most of regions except Siberia, Russia and Ukraine, the injection height from the PRM (i.e. the mean height of maximum injection) is above the injection height from IS4FIRES. The main biomass burning regions: Brazil, Africa North and South of the Equator, SE Asia and Australia, are prominent on all the plots, with injection heights that are higher than other regions such as China, India and Central America. The two
algorithms estimate the higher injection heights over Central Australia, for a limited number of fires however (see Figure 2).

The global statistics from Figure 5 are detailed in table 1 for the regions of interest defined in Kaiser et al. (2012) and shown in Figure 1. In addition to the average, the 1st, 5th (median) and 9th deciles are given for all regions. The mean of the PBL height diagnostic from the ECMWF model for grid points that include fires only is also given. The average of the PRM injection heights is globally more than 150 m higher than the IS4FIRES injection height. The regional variability is high, with
Tropical Asia and Europe showing the lowest injection heights on average for the two algorithms, and Australia the highest. The intra-regional variability is much larger for the PRM than for IS4FIRES injection heights: the lowest plumes are much lower with the PRM, and the highest plume much higher as well. This was also shown by Paugam et al. (2015b) : the PRM seems more able to estimate the higher injection heights. The 9th decile is on average 650 m higher for the PRM as compared to IS4FIRES. Maximum values of injection heights can reach 7 or 8km on occasion with the PRM while they very seldom
reach 4 km with IS4FIRES.

This table also confirms that there is a significant statistical link between PBL height and injection heights, more marked with the PRM. The PBL height diagnostic from ECMWF for grid cells with fires show low values for Tropical Asia. This is probably because most of fires occur quite close to the sea in this region, which comprises mainly Indonesia. The spatial interpolation may thus include PBL height values from over the sea, which are lower than over land because of lower day-time





heat flux from the surface. Besides this region, the regions that display the lowest and the highest averages injection heights with the PRM are also the regions with lowest and highest PBL heights at grid cells with fires. Comparing the 9th decile with PBL height shows that for all regions, the 9th decile of PRM injection heights is significantly above the PBL height diagnostic. This means that with the PRM and for all regions, more than 10% of fires release their constituents mainly in the

free atmosphere. For the IS4FIRES parameterization, this is true only for some specific regions: Australia, South America, North Asia and Tropical Asia. For the most active biomass burning region globally, Africa, North and South of the Equator, the difference between the PRM and IS4FIRES injection heights is significant in terms of the number of fires that release their emissions above the PBL.

This greater spatial and day-to-day variability of the PRM as compared to IS4FIRES is also apparent in Figure 6, which

shows global and regional density plots of median injection heights from the PRM versus IS4FIRES. The linear fit between the two sets indicate that for all regions, high median injection heights are higher with the PRM while low median injection heights are also lower. For some regions such as Europe and North Hemisphere Africa, the median injection height from the PRM is below 500 m in many cases. For these two regions, the mean and median of injection heights are significantly lower with the PRM than with IS4FIRES; which shows that this has a strong impact on regional statistics. This plot also highlights

the regional variability, which is again more marked for the PRM as compared to IS4FIRES injection heights.

### 3.3 Validation against MISR observations of plume top heights

In this section, the newly computed injection heights are compared against the MPHP2 dataset of injection heights derived from MISR observations. By taking into account only the plumes with good and fair retrieval quality, the verification dataset is composed of 13454 injection heights. When collocating these observations against the non-null injection heights computed

from Terra-based fire observations in GFAS, the sample size is reduced to 4182, or 31% of the initial verification dataset. This figure is comparable to the ratio of the global number of grid cells with non-null injection heights over the number of non-null FRP grid cells.

#### 3.3.1 Global scores

Figure 7 presents the global injection height distribution for the MPHP2 dataset as well as for the PRM and IS4FIRES plume

heights. The IS4FIRES injection heights are more often closer to the MPHP2 values, especially for the values that occur the most frequently, between 800 and 1500 m circa. The PRM shows a significant overestimation of the frequency of injection heights below 500 m; this is especially apparent for very low injection heights (200 m). The IS4FIRES parameterization on the other hand underestimates the number of plumes with injection heights below 1000 m. For injection heights above 2500 m, i.e. in the upper tail of the vertical distribution, the PRM gives a frequency that is closer to observations. These plumes constitute

a minority of observed plumes, but they are particularly important in terms of atmospheric composition they are particularly large and they are subject to long-range transport in the free troposphere.

Figure 8 present global density plots of MPHP2 injection heights versus IS4FIRES and PRM injection heights from GFAS. Table 2 presents the global scores and complement Figure 8. The IS4FIRES injection heights show a small positive bias; this





bias is especially marked for observed injection heights that are below 1000 m. On the other hand, for MPHP2 values above 3 km, the IS4FIRES injection heights are often significantly underestimated, by more than 1000 m in a number of cases. This shortfall of the IS4FIRES parameterization was already noted in Sofiev et al. (2012) and Veira et al. (2015a). For MPHP2 plumes between 1000 and 2000 m, which represent a majority of observations, the IS4FIRES injection heights are often within

500 m of the observations.

The larger variability and the more important positive bias of the PRM injection heights are also apparent in Figure 8 and table 2. A significant number of plumes are forecasted with injection heights at 200 m, while observation for these plumes range from 500 to 2500 m. This particular subset brings a significant degradation of the RMSE of the PRM, which is nearly two times as important as the RMSE of IS4FIRES injection heights. The PRM is more able to estimate larger injection heights

as compare to the IS4FIRES parameterization, and a larger fraction of the observed injection heights above 3 km is well forecasted. However, the majority of injection heights estimated by the PRM to be larger than 3 km correspond to MPHP2 values well below 2500 m.

### 3.3.2   Regional scores

Global and regional scores are summed up in table 2. The proportion of estimated plumes that fall within 500 m of the

observations is also displayed. Globally and for all regions, 55 to 65% of injection heights from IS4FIRES fall within 500 m of the observations against 40 to 50% for PRM injection heights. Correlation between estimations and observations is also higher for IS4FIRES's injection heights, with values ranging from 0.4 to 0.55 for the correlation coefficient, against 0.3 to 0.4 for PRM injection heights. The figures for IS4FIRES injection heights are close to the results of Sofiev et al. (2012); the fraction of plumes that fall within 500 m of the observations being a bit lower.

The lower scores of the PRM injection heights are associated with a much larger RMSE and bias, globally and for all regions. The two algorithms show a correlation in both bias and RMSE: The regions with maximum and minimum bias and error are the same for both algorithms, with larger values for the PRM. This is because the errors are larger for higher injection heights in both calculations.

### 3.3.3   Scores per biome type

The MPHP2 dataset also include the MODIS land cover type product following the International Geosphere-Biosphere Program (IGBP) classification (Channan et al. (2014)). Scores for each IGBP biome type are shown in table 3. They help understand the strengths and weaknesses of each algorithm. The PRM shows a large bias of 664m for evergreen broad-leaf forests, and a bias of more than 350 m for savannas and woody savannas. This could come from the fact that the parameters of the PRM were derived by optimization against a small number of plumes in North America (Paugam et al. (2015b)), which may

not at all include plumes over this kind of land cover. For these biome types, which represent more than half of all the MPHP2 dataset, the RMSE is also very large for the PRM.

The RMSE and bias are both significantly lower for IS4FIRES injection heights, for all biome types. The variability depending on the biome type is also lower for IS4FIRES. For both injection heights, the best results are achieved on fires occurring in





deciduous needle-leaf forests areas, i.e. boreal fires. For the PRM, optimization of the parameters was done on fires occurring just over such land covers.

### 3.3.4 Discussion

There are three sources of errors in the injection height estimates that were assessed in this study (besides possible errors on the verification dataset):

- Uncertainties on the MODIS satellite products that are used as input,

- Uncertainties on the ECMWF profiles that are used to provide information on the environment,

- And finally, simplifications and/or processes not represented in the algorithms.

These three sources of errors all play a role in the scores. MODIS FRP is often underestimated (Kahn et al. (2008), Schroeder et al. (2014)). This underestimation seems to be more significant for large fires, because of the opacity effect whereby the smoke fire hinders the remote detection of thermal anomalies at the surface. This impacts particularly the IS4FIRES parameterization which has a stronger statistical link to FRP. Veira et al. (2015a) improved scores of another version of IS4FIRES against the MPHP dataset by applying an empirical FRP correction for plumes higher than a threshold height. This hasn't been tried yet in this study; however this modification may also probably increase the small positive bias against MPHP2 injection heights of the IS4FIRES injection heights. Tests were carried out using the accumulated FRP of the fire cluster instead of the cluster maximum FRP as an input for the IS4FIRES parameterization. This didn't have much impact on the output and on the scores, despite a much larger input value for FRP of large fires.

Using additional satellite products, especially from geostationary satellites, can only improve the quality of the injection heights and of the GFAS products as a whole. This would also allow to describe the diurnal cycle of fires and injection heights. Work is going on to combine the accuracy of low orbit observations with the better time resolution of geostationary products in GFAS (Andela et al. (2015)).

The two algorithms are very dependent on the atmospheric profiles provided by the ECWMF. The diagnosed PBL height is of special important since plumes often reach the top of the PBL without breaking through to the free troposphere. Even if the ECMWF diagnostic appears to be of good quality generally (see the studies of Palm et al. (2005) and Flentje (2014)), possible errors will have a large impact on injection height estimates from both algorithms. In particular, it seems that the numerous cases where the PRM estimates very small injection heights (200-300 m) while the MPHP2 dataset indicated injection heights from 1000 to 2000 m correspond to cases where the PBL height according to ECMWF was underestimated. This led to a notable degradation of the scores of the PRM injection heights in GFAS. In some cases, especially over Indonesia, some land-sea mask representativity issues arose for the ECMWF environment profiles, caused by the coarse 1x1°grid that was used. The bad scores of the PRM over Indonesia were improved when using a better resolution for the atmospheric environment. The extra computing cost of increasing the resolution of the environment is however too great to consider this option for running GFAS in NRT mode or to produce climatology.





Also, the fact that the PBL height diagnosis from ECMWF was used in IS4FIRES instead of the PBL height formulations used in Sofiev et al. (2012) and Sofiev et al. (2013) could be a source of error for this algorithm.

The PRM is a much more complex algorithm than IS4FIRES, using an estimate of fire size as an additional input, and taking into account wind drag, plume microphysics, entrainment and detrainment in its parameterization. This increased complexity makes it more able to estimate large injection heights; however it also means that it is less robust than IS4FIRES in the sense that it is subject to more sources of uncertainty, notably on fire size. The fact that the 6 parameters that are needed in its system of equations were fitted from a small number of plumes may have degraded its results, especially for the land cover types that were not taken into account in Paugam et al. (2015b).

Additionally for the PRM, the fact that a whole detrainment profile is translated into only 3 parameters (mean height of maximum of injection, top of the plume, bottom of the plume) is an additional source of error.

## 4   Using GFAS injection heights in atmospheric composition modeling: examples

This chapter presents two case studies for the use of assimilated injection heights from GFAS in atmospheric composition modeling: the SAMBBA campaign which took place in September 2012 in Brazil, and a series of large fire events in the Western U.S. in August 2013. The biomass burning emissions and injection heights are used in the Composition-Integrated Forecasting System (C-IFS) model.

### 4.1   The C-IFS forecasting system

The C-IFS global atmospheric composition model is the flagship model of the Copernicus Atmosphere Monitoring Service (CAMS). CAMS is a European Union funded operational service that operates an assimilation and forecasting system for monitoring aerosols, greenhouse gases and reactive gases, using satellite observations and a combination of global and regional models (Hollingsworth et al., 2008; Peuch and Engelen, 2012). The meteorological component of C-IFS is based on the ECWMF operational model, the Integrated Forecast System (IFS). The trace gases component of C-IFS is described in Flemming et al. (2015).

Aerosols are forecasted within the C-IFS global system by a forward model (Morcrette et al. (2009), based on earlier work by Reddy et al. (2005) and Boucher et al. (2002)) that uses five species: dust, sea-salt, black carbon, organic matter and sulfates. GFAS provides the biomass burning emissions of black carbon, organic matter and sulfates. In this chapter the sum of black carbon and organic matter is hereafter denoted as "biomass burning aerosols". C-IFS provides and uses aerosol analysis by assimilating total Aerosol Optical Depth (AOD) observations from MODIS in a 4D-Var assimilation algorithm, as described in Benedetti et al. (2009).

C-IFS used to emit biomass burning aerosols at surface. For this study, the model was modified to take into account the biomass burning emissions at the level of the PRM mean height of maximum injection from the GFAS. A criterion on the PBL height diagnostic was added to prevent emitting biomass burning aerosols too high at night. When the PBL height is above 500 m, the emissions are distributed in the three model levels that are surrounding the injection height provided by the PRM.



Since the PRM appears to have more ability to estimate high injection heights, which are associated with large fires, the PRM was chosen for as a first implementation. The following case studies will help to further evaluate the performance of the PRM for high injection heights. However, the IS4FIRES injection heights will be used in an upcoming test, since they show better scores overall in comparison with the MPHP2 dataset.

## 4.2 SAMBBA field campaign

The South AMerican Biomass Burning Analysis (SAMBBA) campaign was an intensive field campaign which took place in September-October 2012 in Brazil, aimed at investigating the properties of biomass burning aerosols over the Amazon basin. The main biomass burning season occurs there during July-October, when deforestation fires and agricultural burning are frequent. Marenco et al. (2015) analyzed lidar observations of smoke aerosols from six flights of the Facility for Airborne Atmospheric Measurements (FAAM) BAe-146 research aircraft. The ground tracks of the considered flights are shown in Figure 9 and table 4, reproduced from Marenco et al. (2015). The flights took place during intense biomass burning episodes in the East of the Amazon Basin. Figure 9 shows the PRM top of the plume and mean height of maximum injection from GFAS for the 16 to 29 September 2012 period. The first period, 16-22 September, is characterized by intense fires, with mean heights of maximum injection often reaching more than 4 km between between 10 and 15°S and 45-55 °W, and top of the plume estimated at more than 5 km for some fires in this box.

The cross-sections of the aerosol extinction coefficient from the lidar observations for the six considered research flights are presented in Figure 10. All of the six flights show enhanced extinction due to biomass burning, with smoke layers lying at altitudes varying from 2 to 4 km. Extinction profiles from several flights (B733 and B741 especially) show two or more distinct aerosol layers, which may originate from distinct fires. The elevation of the observed aerosol layers is close to the mean heights of maximum injection provided by GFAS in the region, i.e. between 3 and 4 km generally.

As detailed in Marenco et al. (2015), the observed profiles of aerosol extinction were compared against model predictions from the Met Office United Model (MetUM) and C-IFS. The results from the MetUM are presented and analyzed in Marenco et al. (2015). Figure 11 shows the simulated extinction profiles from C-IFS along the flight track, with biomass burning aerosols injected at surface and at the PRM mean height of maximum injection provided by GFAS.

Except for flight B741, the aerosol extinction is larger when the biomass burning aerosol are injected at an altitude. Comparing to observations, using the PRM injection height from GFAS seems to bring an improvement in the forecasts of the aerosol extinction profiles for flights B742, B743 and also B746. For these flights, the observed layers of aerosols at 4 km (B742), 2 km (B743) and 2.5 km (B743) are better represented when using injection heights while they are either nearly absent (B742) or underestimated (B743 and B744) when emitting at the surface. For flight B742, the modified model is able to forecast the two aerosol layers that were observed. For flights B734 and B741, there is no discernible improvement or degradation while for flight B733, the use of injection heights increased the simulated aerosol burden in the smoke plume between 2 and 3 km beyond the observed values.





### 4.3 SEAC4RS field campaign

In the framework of the Studies of Emissions, Atmospheric Composition, Clouds and Climate Coupling by Regional Surveys (SEAC4RS) field campaign, the National Aeronautics and Space Administration (NASA) operates a DC-8 aircraft to sample the smoke plumes from fires in continental United States. Aerosol extinction was remotely sensed using a combined High
Spectral Resolution Lidar (HSRL, Hair et al. (2008)) at 532nm and Ozone Differential Absorption Lidar at 290nm (DIAL, Browell et al. (1983)). Measurements here focus on the aerosol measurements from the HSRL. These observations were used for the studies of the Rim Fire, which occurred in late August 2013 in California (Peterson et al., 2015) and were used to derive fire emissions using inversion techniques (Saide et al. (2015)).

Here, we focus on a single flight from the DC-8 on 19th of August 2013. The track of the flight and the observed height of
the mixed layer are shown in Figure 12. The cross-section of the aerosol extinction at 532nm observed by the DIAL-HSRL are presented in Figure 13a. Its most distinct feature is an elevated biomass burning aerosol layer, between 4 and 6 km high, from 35 to 45°N, caused by large fires raging in the United States Pacific Northwest. Most of the aerosol burden lies above the top of the PBL, which was between 1 and 1.5 km (Figure 12).

The C-IFS forecasting system was run with biomass burning aerosols emitted at surface (13b) and at the PRM mean height
of maximum injection from GFAS (Figure 13c). The impact of using injection heights is shown on Figure 13d. The extinction resulting from the biomass burning is higher throughout the sampling regions when the injections heights were provided by GFAS as compared to emitting at the surface. The difference is especially important for altitudes between 2 and 6 km at 16:00 (35 °N), around 2 km between 18:00 and 19:00 (43-45 °N) and around 4 km between 20:00 and 22:00 (35-40 °N). The simulated values from C-IFS with injection at surface underestimated aerosol extinction at altitudes higher than 3-4 km and
sometimes overestimated extinction close to the surface. When using injection heights from GFAS, the smoke layer at higher altitudes is better represented, but the overestimation close to the surface is also larger. The elevated layer of aerosols around 6 km is not captured by either simulation.

### 4.4 Injection height trend analysis

Figure 14 presents the monthly and yearly averages of the median of IS4FIRES and PRM injection heights from GFAS. The
seasonal cycle is more or less marked depending on the regions; regions close to the equator show the smallest amplitude while boreal regions show a very marked amplitude. During winter, the number of GFAS grid cells with positive injection heights, shown in figure 15, is very small there. The global trend is a 3 m per year increase; which is rather small. The regional trends range between -11 m and 4 m per year, with the largest year-on-year increase occurring in North Hemisphere Africa. The maximum number of grid-cells with positive injection heights in Tropical Asia occurred during the El Nino-
Southern Oscillation (ENSO) year 2006, the most intense for the period considered in this study. This was associated with larger emissions of biomass burning aerosols and CO there (Inness et al. (2015)).

The global and regional trends of the median PRM injection height are much more important than for the IS4FIRES parameterization . The global trend is positive, at 16 m per year. The global evolution is driven mainly the the marked yearly increase



of injection heights in North Hemisphere Africa (23 m per year) and South Hemisphere Africa (10 m per year). North America also show a positive trend in mean injection height (11 m per year), but it contributes far less to the total trend since there are far fewer fires as in North- and South-Hemisphere Africa (see Figure 15).

The trend in number of injection heights is negative, with important contributions from North-Hemisphere Africa and South America. A small positive trend in Australia is associated with two active fire seasons there in 2011 and 2012. This decrease in number of injection heights corresponds to a year-on-year decrease of number of pixels with positive FRP in GFAS (not shown). It is also associated with a decrease of regionally averaged FRP in South America, but not in North-Hemisphere Africa (not shown). This decrease can be caused either by one or two of the MODIS sensors being less sensitive to small fires over the years, or by a real downward trend in number of fires. For fires in Brazil, the number of deforestation-induced fires are reduced in recent years (Wiedinmyer (2015)) and has been noted by FINN, GFAS as well as GFED4 Giglio et al. (2013). A downward trend in burnt area in North-Hemisphere Africa was also noted in GFED4 for recent years (Andela et al., 2014), which seems to confirm our findings.

In South America, a significant decrease in number of fires, injection heights, average FRP, and no trend in the median injection height from either the PRM or IS4FIRES indicate that the decrease in fire occurrence concerns small and large injection heights equally. In North Hemisphere Africa, the decrease in number of fires and injection heights is associated with a significant increase in median injection heights, more marked with the PRM, and with constant average FRP over the years. This means that small injection heights (associated with less intense fires) are more affected by this decrease. Globally, the decrease in fire occurrence seems to affect fires with smaller injection heights.

Figure 16 shows time series of median FRP and of the PBL height diagnostic from the operational ECMWF model, over grid cells in which fires are active. The number of fire pixels (not shown) follows a similar trend as the number of pixels with a positive injection height. This tendency is associated with a smaller decrease in median FRP: the yearly rate of decrease is around two times smaller for FRP as compared to number of fires. There is no discernible trend for the PBL height above fire locations, which displays a high yearly variability.

The same figure also shows the correlation between monthly averages of FRP and PBL height over fires versus PRM and IS4FIRES injection heights. The correlation is higher with PBL height for the two algorithms. Monthly averaged IS4FIRES injection heights are significantly correlated with both FRP and PBL height. This is clearly a large-scale phenomenon, since when comparing at the scale of a MODIS pixel (see Figure 2 and 3), the correlation is very small. As a consequence, it appears easier to parameterize injection heights as a function of FRP and PBL height for larger scales than for individual fires. This means that the added values of the GFAS database lies in the daily time scale and its spatial resolution of 0.1°.

## 5   Conclusions

Two existing algorithms estimating fire injection heights have been embedded into the GFAS system: the IS4FIRES parameterization and the Plume Rise Model. The new system, GFAS, provides daily global fire emissions and injection heights at a resolution of 0.1 °. It uses as input FRP from MODIS and atmospheric profiles from the operational ECMWF model. GFAS





has been run for the 1/1/2003 to 1/1/2015 period, and is now running in near real-time to provide biomass burning emissions for the C-IFS model. The GFAS climatology of fire emissions has thus been extended to include injection heights from two very different algorithms.

This dataset is publicly available at the CAMS-GFAS dataserver at http://apps.ecmwf.int/datasets/data/cams-gfas. It can

serve a variety of purposes, from the study of past fires to global atmospheric composition monitoring within the CAMS project for example. The fire emissions and injection heights from GFAS can also provide inputs within the framework of model intercomparison projects such as the Aerosol Comparisons between Observations and Models (AEROCOM) project for example.

Injection heights from GFAS were assessed against the MPHP2 dataset. The IS4FIRES injection heights show better scores

globally and locally, but the PRM seems better at estimating large injection heights and at reproducing the variability of injection heights. Several causes of error possibly degrade the scores: the PBL height diagnostic from ECMWF that is used in the two algorithms, and errors or underestimations of MODIS FRP. The two datasets can also be used together in an optimal combination.

GFAS showed a trend towards a decreasing number of grid-cells with fires and positive injection heights. This global trend is

driven by regional trends in South America and North hemisphere Africa. Comparing the trends in median injection heights and FRP yields the conclusion that whereas the number of fires decrease in the two regions (this is confirmed by other inventories), this decrease affects all kind of fires in South America, and more frequently smaller fires in North hemisphere Africa.

The use of injection heights from GFAS for biomass burning emissions has been implemented in the global atmospheric composition model C-IFS, and it was shown that for two particular fire situation, they bring the model profiles of aerosol

extinction generally closer to observations. The forecast of high (i.e. above 4 km of altitude) plumes in particular seems to be improved by using injection heights in the C-IFS system.

The GFAS system currently uses FRP observations from MODIS only. It has recently been upgraded (Remy and Kaiser (2014)) so as to be able to assimilate observations from other satellites. These might eventually also be used in the two injection height algorithms of GFAS, thus enhancing the robustness of the system.

*Acknowledgements.* The authors wish to thank NASA for providing the MODIS data, and Charles Ichoku for providing Figure 1 reproduced from Ichoku and Ellison (2014).

The IS4FIRES algorithm was developed within the IS4FIRES project of the Academy of Finland.

Airborne data from the SAMBBA field campaign were obtained using the BAe-146-301 Atmospheric Research Aircraft (ARA) flown by Directflight Ltd and managed by the Facility for Airborne Atmospheric Measurements (FAAM), which is a joint entity of the Natural

Environment Research Council (NERC) and the Met Office.

Support for the DIAL/HSRL operations and measurements during the SEAC4RS field campaign was provided by NASA Headquarter's Tropospheric Chemistry, Radiation Sciences, and Upper Atmospheric Research Programs. Support for the development and addition of the HSRL capability was provided by the NASA Airborne Instrument Technology Transition Program.

This research was supported by the EU Seventh Research Framework Programme (MACC-III project, contract number 283576).



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


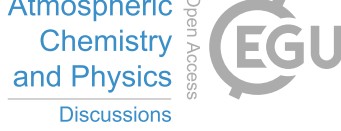

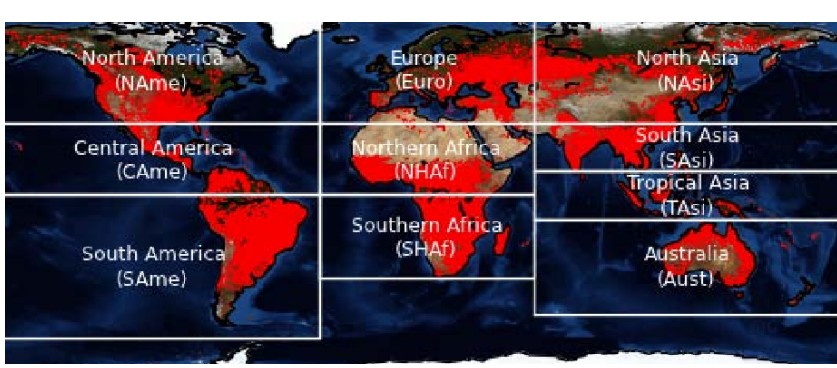

**Figure 1.** Regions of interest as defined in Kaiser et al. (2012).





**Figure 2.** Global average for the 1/1/2003 - 1/1/2015 period of yearly number of non-null values for injection heights (top) and Fire Radiative Power (middle) in GFAS. The ratio between the two is presented at the bottom.





**Figure 3.** Comparison of the ECMWF boundary layer height diagnostic against raw height of maximum injection from the PRM (top) and from the IS4FIRES algorithm (middle) for the 1/6/2013 - 1/8/2013 period. Bottom, comparison of the raw height of maximum injection from the PRM against the injection height from IS4FIRES. Observations used are from Terra (left) and Aqua (right).







**Figure 4.** Comparison of observed maximum FRP from Aqua (top) and fire area estimated by the Dozier algorithm using Aqua observations (bottom) against raw height of maximum injection from the PRM (left) and injection height from IS4FIRES (right), for the 1/6/2013 - 1/8/2013 period.





**Figure 5.** Global average for the 1/1/2003 - 1/1/2015 period of daily injection heights from GFAS. Top of the plume (top) and mean height of maximum injection (middle) from the PRM, and injection height estimated with the IS4FIRES parameterization (bottom). The averages were computed taking into account only non-null values.





**Figure 6.** Density plots of global and regional comparison of the median of IS4FIRES's injection height against the median of the PRM mean height of maximum injection for the period 1/1/2003 - 1/1/2015. The dashed red line indicate the best linear fit.





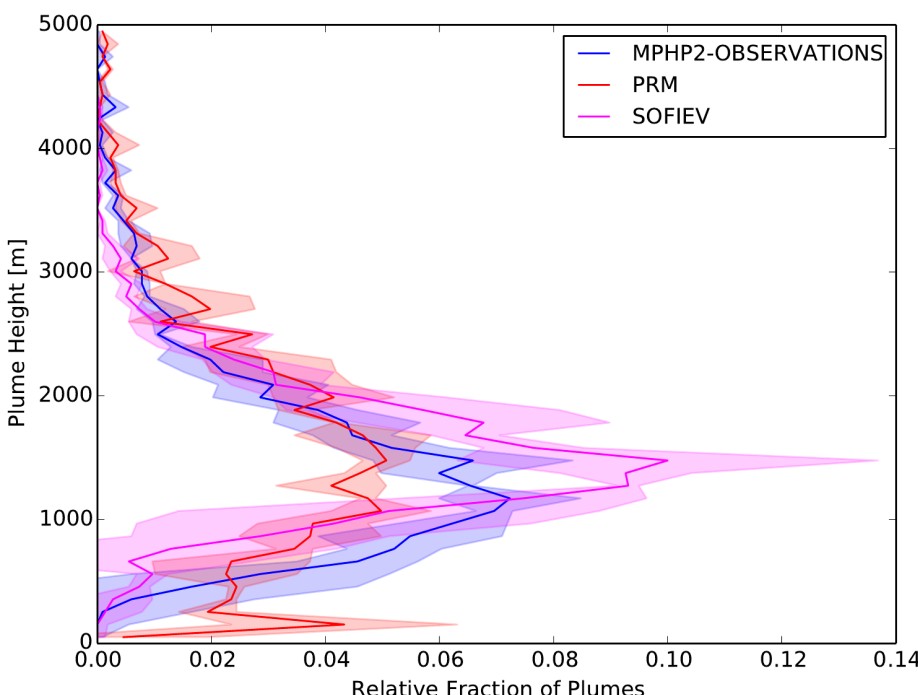

**Figure 7.** Global distribution of injection height from IS4FIRES and mean height of maximum injection from the PRM, and of MPHP2 observations for 2008. Shading represents uncertainties of 200 m in the plume height observations and parametrizations. The sum over the vertical of the relative fractions is 1 for all curves.





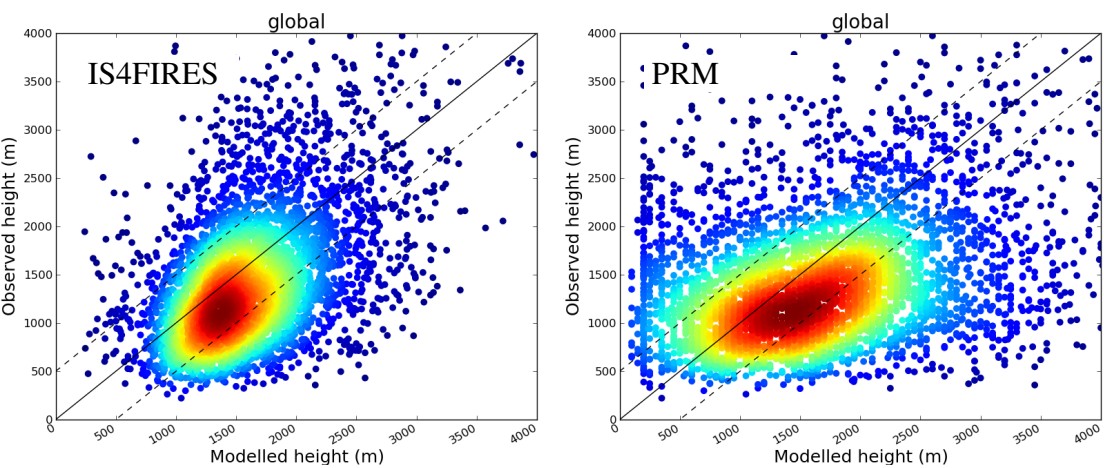

**Figure 8.** Density plots of global comparison of the MPHP2 observations of plume height in 2008 against the IS4FIRES (left) and the PRM (right) injection heights of GFAS. MPHP2 observations with "good" and "fair" quality flags only were used. The dashed lines delineate the area in which the GFAS injection heights are within 500 m of observations.





**Figure 9.** Left, PRM mean height of maximum injection from GFAS, accumulated values from 16-22 September 2012 (top) and 23-29 September 2012 (middle). Right, PRM top of the plume from GFAS, accumulated values from 16-22 September 2012 (top) and 23-29 September 2012 (middle). At the bottom, ground tracks of the six research flights of the SAMBBA field campaign listed in Table 4.





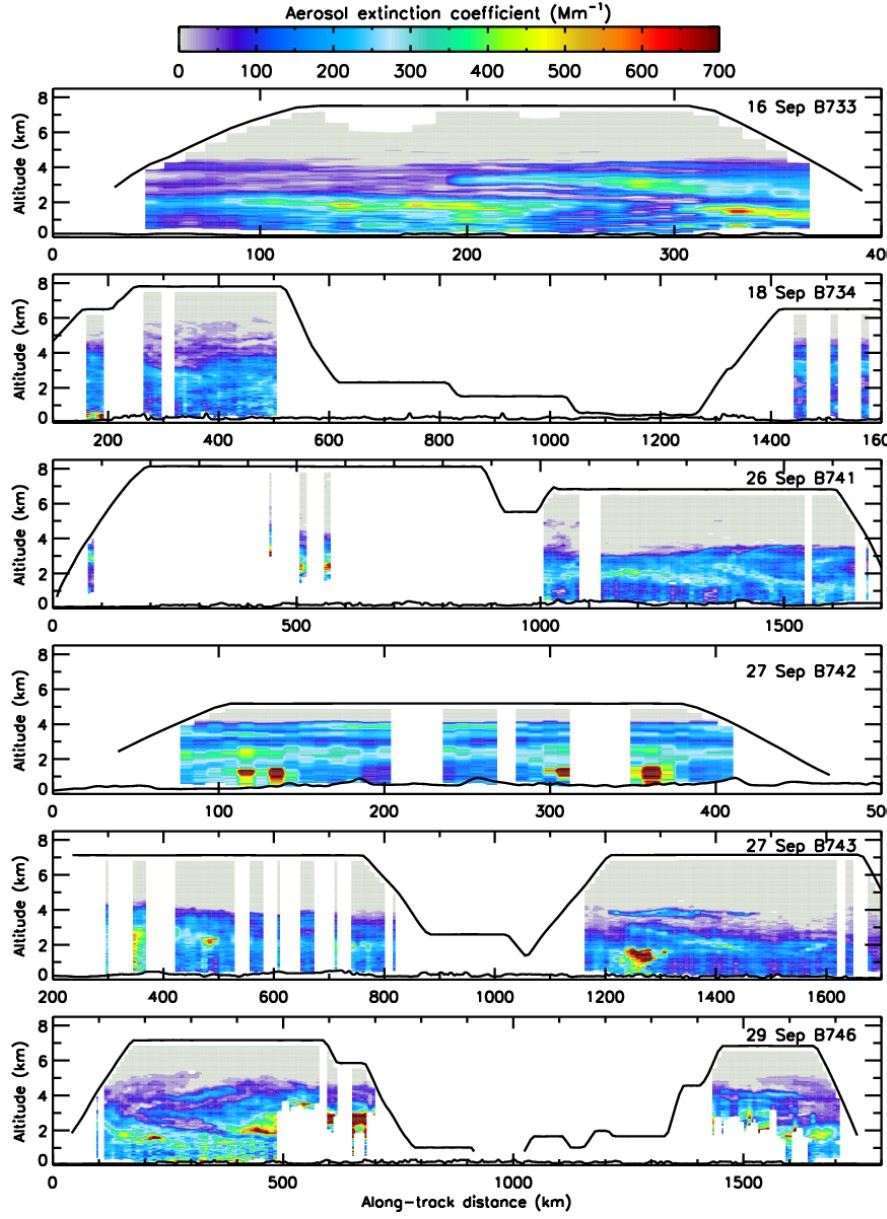

**Figure 10.** Cross-sections of the aerosol extinction at 355nm coefficient determined from the lidar for the six research flights with a 1-minute integration time. The black lines indicate the aircraft altitude and the surface elevation from a digital elevation model, respectively.



**Figure 11.** Cross-sections of the aerosol extinction coefficient at 532nm estimated from the C-IFS along the tracks of the six research flights. Biomass burning aerosols are emitted at the surface (left) and at the PRM mean height of maximum injection provided by GFAS (right).





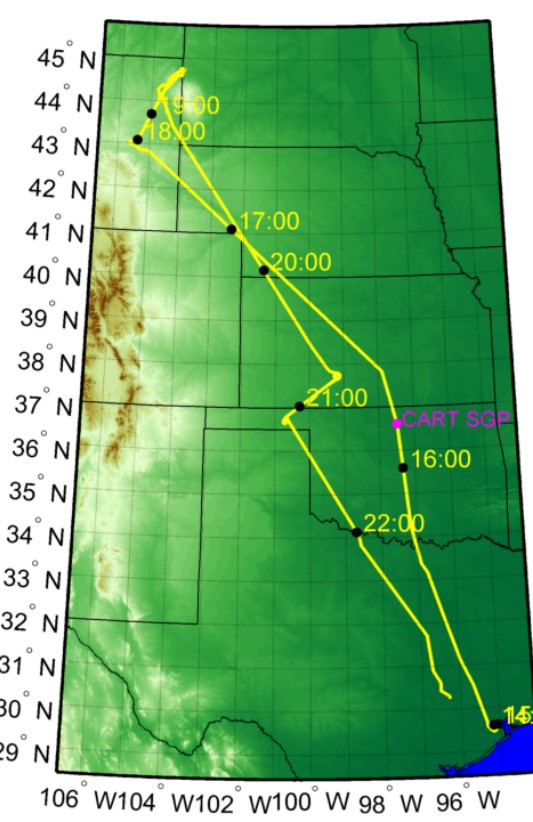

**Figure 12.** Ground tracks for the 19th August 2013 research flight of the SEAC4RS field campaign, in southern U.S.A.. CART SGP stands for the Cloud and Radiation Testbed (CART) Southern Great Plains (SGP) site.





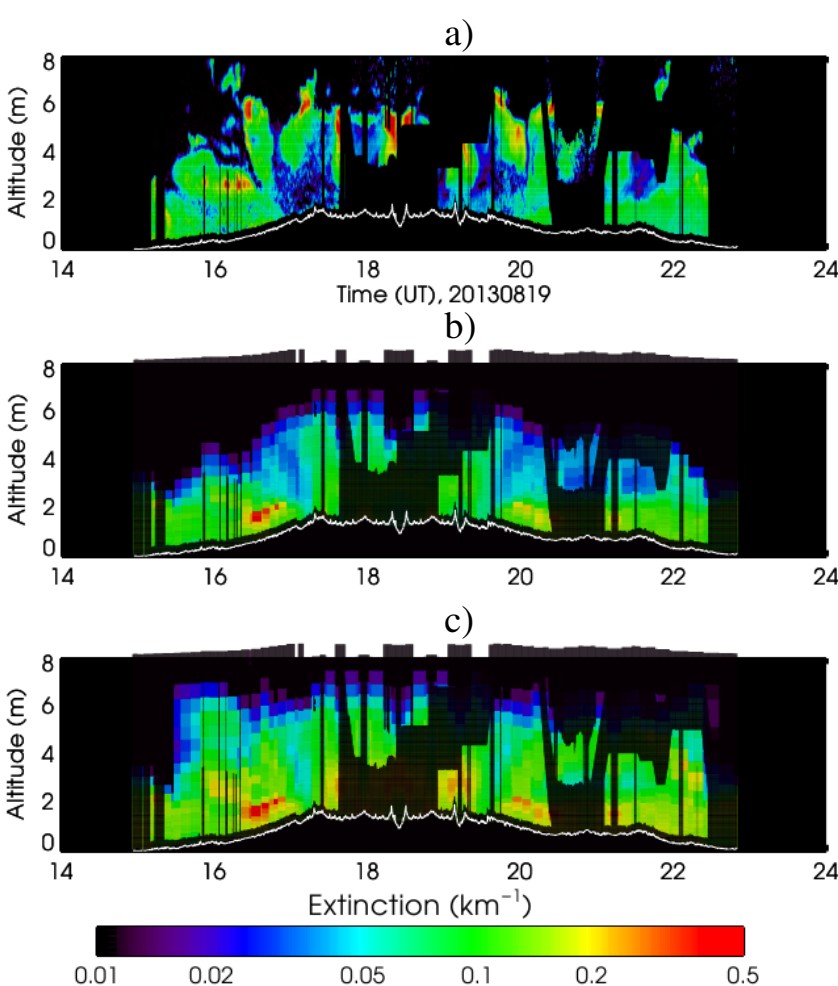

**Figure 13.** Cross-sections of the aerosol extinction at 532nm; a), observed along the tracks of the research flights of SEAC4RS of 19/8/2013; b) C-IFS forecast with biomass burning aerosols emitted at surface and c) at the PRM mean height of maximum injection provided by GFAS.



**Figure 14.** Global and selected regional monthly (red) and yearly (blue) averages of the median of IS4FIRES (left) and PRM (right) plume heights from GFAS. The green dashed line indicate the linear fit for the period 1/1/2003 - 1/1/2015.





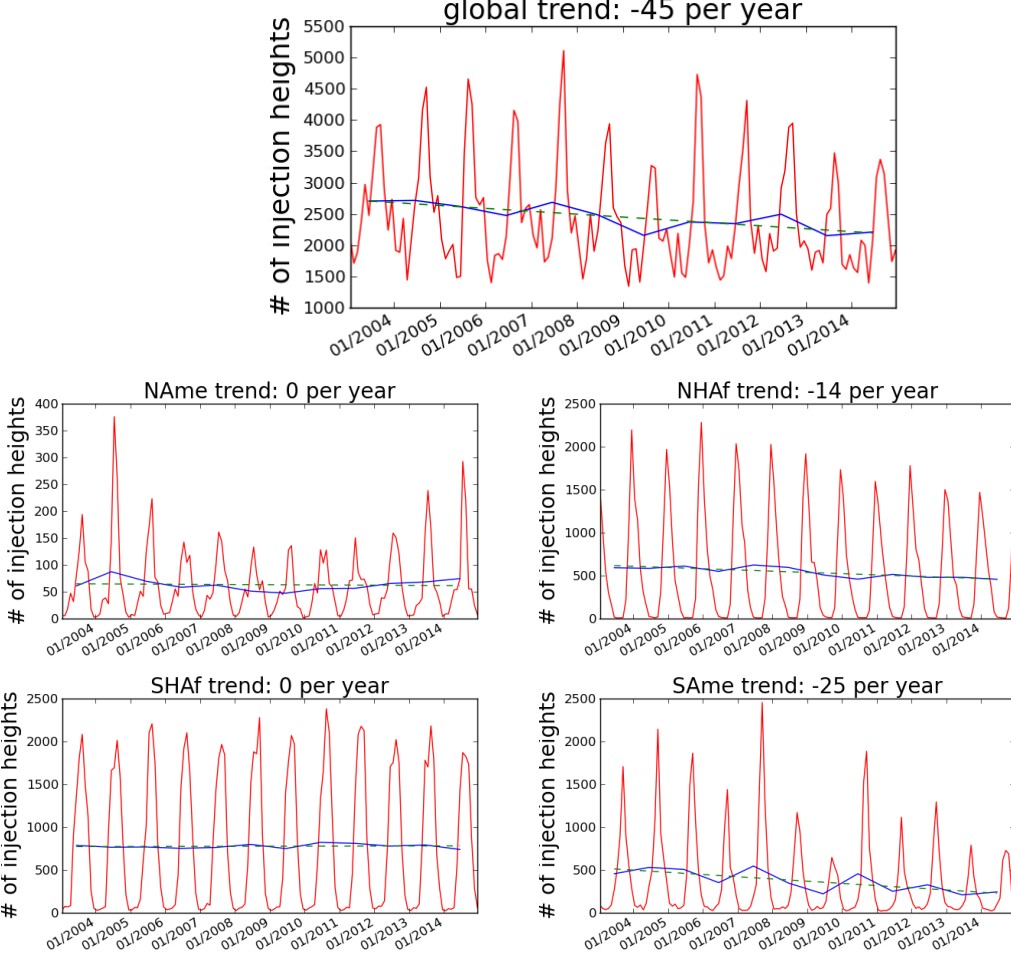

**Figure 15.** Global and regional monthly (red) and yearly (blue) averages of the number of the daily grid cells with non-null injection heights from GFAS. The green dashed line indicate the linear fit for the period 1/1/2003 - 1/1/2015.





**Figure 16.** On top, Global monthly (red) and yearly (blue) averages of the median FRP (left) and of boundary layer height above fire grid cells (right). Below, density plot of monthly averaged global FRP (left) and PBL height above fires (right) against monthly averaged PRM injection heights (middle) and IS4FIRES injection heights (bottom).





**Table 1.** Statistical analysis of the IS4FIRES and the PRM plume height in GFAS over the period 1/1/2003 - 1/1/2015, together with the mean of the PBL height diagnostic from the operational ECMWF model at the points where there were fires. All values are in meters and computed for the same fires. Only the non-null values were used in the computation of the statistics. "IS4" stands for "IS4FIRES Parameterization", "PRM" for "Plume Rise Model"

| Region name | mean-IS4/PRM | mean PBL h. | 1st decile-IS4/PRM | 5th decile-IS4/PRM | 9th decile-IS4/PRM |
| --- | --- | --- | --- | --- | --- |
| Global | 1377 / 1536 | 1634 | 792 / 431 | 1316 / 1460 | 2026 / 2680 |
| Australia | 1522 / 1570 | 1955 | 889 / 465 | 1465 / 1567 | 2218 / 2616 |
| Central America | 1173 / 1164 | 1620 | 818 / 388 | 1129 / 1092 | 1586 / 1963 |
| Europe | 1021 / 836 | 1585 | 591 / 245 | 985 / 740 | 1490 / 1550 |
| North America | 1361 / 1193 | 1611 | 858 / 396 | 1398 / 1138 | 1928 / 2018 |
| North Asia | 1179 / 1022 | 1493 | 741 / 320 | 1144 / 966 | 1656 / 1768 |
| N. Hem. Africa | 1168 / 1309 | 2054 | 685 / 425 | 1167 / 1184 | 1771 / 2333 |
| S. Hem. Africa | 1272 / 1424 | 2321 | 801 / 529 | 1209 / 1359 | 1814 / 2364 |
| South America | 1316 / 1375 | 1809 | 829 / 371 | 1255 / 1305 | 1882 / 2406 |
| South Asia | 1105 / 1147 | 1591 | 788 / 466 | 1066 / 1096 | 1472 / 1871 |
| Tropical Asia | 983 / 1092 | 1079 | 697 / 404 | 954 / 916 | 1306 / 1970 |





**Table 2.** Scores of IS4FIRES / PRM plume height against the MPHP2 dataset for the year 2008, for each regions defined in figure 1. The "% inside" score correspond to the relative fraction of estimated plumes that are within 500 m of the MPHP2 observations

| Region name | sample size | RMSE-IS4/PRM (m) | bias-IS4/PRM (m) | mean-IS4/PRM (m) | R-IS4/PRM | % inside-IS4/PRM |
|---|---|---|---|---|---|---|
| Global | 4182 | 533 / 955 | 144 / 239 | 1619 / 1714 | 0.45 / 0.31 | 57 / 44 |
| Australia | 306 | 652 / 863 | 120 / 186 | 1555 / 1620 | 0.55 / 0.44 | 57 / 50 |
| Central America | 151 | 426 / 684 | 117 / 43 | 1453 / 1380 | 0.22 / 0.10 | 63 / 50 |
| North America | 251 | 632 / 1071 | -69 / -110 | 1832 / 1791 | 0.59 / 0.37 | 55 / 42 |
| Europe | 196 | 489 / 879 | -133 / -96 | 1596 / 1634 | 0.41 / 0.33 | 53 / 45 |
| North Asia | 997 | 497 / 852 | -13 / -73 | 1626 / 1567 | 0.47 / 0.32 | 64 / 49 |
| North Hem. Africa | 836 | 486 / 1073 | 180 / 401 | 1616 / 1836 | 0.39 / 0.40 | 57 / 44 |
| South Hem. Africa | 825 | 631 / 1010 | 375 / 584 | 1668 / 1876 | 0.42 / 0.38 | 54 / 40 |
| South America | 520 | 556 / 1108 | 273 / 424 | 1585 / 1736 | 0.47 / 0.30 | 48 / 61 |
| South Asia | 90 | 537 / 969 | 26 / 63 | 1307 / 1339 | 0.46 / 0.40 | 65 / 61 |
| Tropical Asia | 11 | 297 / 1670 | -182 / 923 | 1357 / 2462 | 0.14 / 0.13 | 64 / 18 |

**Table 3.** Scores of IS4FIRES / PRM plume height against the MPHP2 dataset for the year 2008, for each IGBP biome type. The "% inside" score correspond to the relative fraction of estimated plumes that are within 500 m of the MPHP2 observations

| IGBP Biome type | sample size | RMSE-IS4/PRM (m) | bias-IS4/PRM (m) | mean-IS4/PRM (m) | R-IS4/PRM | % inside-IS4/PRM |
|---|---|---|---|---|---|---|
| All | 4182 | 533 / 955 | 144 / 239 | 1619 / 1714 | 0.45 / 0.31 | 57 / 44 |
| Evergreen broad-leaf | 248 | 521 / 1222 | 289 / 664 | 1475 / 1850 | 0.42 / 0.29 | 51 / 40 |
| Deciduous needle-leaf | 229 | 428 / 705 | -87 / 17 | 1533 / 1637 | 0.47 / 0.21 | 65 / 48 |
| Mixed forest | 206 | 503 / 816 | -26 / -190 | 1557 / 1394 | 0.40 / 0.21 | 62 / 51 |
| Open shrublands | 318 | 659 / 961 | 61 / 114 | 1725 / 1779 | 0.54 / 0.40 | 58 / 48 |
| Woody savannas | 1040 | 566 / 942 | 272 / 403 | 1596 / 1833 | 0.40 / 0.37 | 58 / 43 |
| Savannas | 1058 | 524 / 1063 | 180 / 376 | 1636 / 1833 | 0.41 / 0.37 | 56 / 43 |
| Grasslands | 316 | 533 / 953 | 47 / -67 | 1811 / 1697 | 0.35 / 0.29 | 51 / 42 |
| Croplands | 239 | 472 / 851 | 73 / 65 | 1566 / 1558 | 0.50 / 0.39 | 56 / 50 |
| Cropland / veg. mosaic | 327 | 444 / 851 | 117 / 65 | 1511 / 1525 | 0.40 / 0.24 | 64 / 48 |





**Table 4.** Research flights of the SAMBBA field campaign considered in this article. Time is UTC.

| Flight | Date | Takeoff | Landing | Latitude | Longitude |
|--------|------|---------|---------|----------|-----------|
| B733 | 16 Sep | Rio Branco, 13:51 | Porto Velho, 14:45 | 8.9-9.8S | 64.5-67.6W |
| B734 | 18 Sep | Porto Velho, 12:05 | Porto Velho, 16:01 | 8.9-11.9S | 61.6-64.4W |
| B741 | 26 Sep | Porto Velho, 12:53 | Palmas, 16:08 | 8.8-10.2S | 48.7-63.9W |
| B742 | 27 Sep | Palmas, 12:52 | Palmas, 16:17 | 10.2-11.5S | 46.8-48.1W |
| B743 | 27 Sep | Palmas, 18:08 | Porto Velho, 21:34 | 9.0-10.2S | 48.4-63.6W |
| B746 | 29 Sep | Porto Velho, 12:54 | Porto Velho, 16:38 | 8.7-9.4S | 58.2-63.7W |