# Peer review of "Two global datasets of daily fire emission injection heights since 2003"

_Atmospheric Chemistry and Physics, 2015_

## Referee Comment (RC1) · Anonymous Referee #1 · 25 Jul 2016

General Comments

Emissions from vegetation fires contribute a major part of the atmospheric aerosol load and contribute to the chemical composition of the atmosphere. A crucial parameter that determines the atmospheric concentrations of gaseous and particulate matter released by fires is the effective source height. The burning vegetation releases heat into the atmosphere producing buoyancy that can lead to local plume rise in the order of several thousand meters. Beside the buoyancy the environmental conditions like static stability and wind shear are major factors that determine the plume rise. Atmospheric models describing the spatial and temporal distribution of gases and aerosols therefore urgently need emission data that beside source strength also need the effective source height or profiles of the emission strength.
The authors describe how the injection heights are provided within the GFAS. They use two fundamental different approaches. They call it the semi-empirical parameterization of Sofiev et al. (2012) and the analytical one-dimensional plume rise model (Paugham et al., 2015). The results are evaluated with plume heights derived from satellite data (MPHP data set). This work is of great significance since the emission heights are a key parameter for aerosol modelling studies. In two case studies the extended GFAS dataset is used to improve the simulated aerosol distribution.

The paper is therefore an important contribution for the modelling community. Unfortunately, in its present form the paper has to be rejected due to the lack of presentation quality. Examples (there would be many things more to add) leading to my overall judgment of the paper are given in the specific comments.

As there are native speakers in the list of authors I would encourage them to check the language style, the grammar, and the wording of the paper.

In the following I will make a detailed suggestion for a revised version and highly encourage the authors to re-submit their important description of the new created input data for the modelling community.

The general structure of the revised manuscript should be:

1 Introduction The introduction needs no subsections. It should summarize the current state in the field and at the end explain what the reader can expect from the current paper.

2 Methodology and data used for validation In this section a sketch is to be included to explain the output of the two plume rise models. It is important to show the reader what injection height, top of the plume, bottom of the plume, mean height of maximum injection, plume height and others are. Symbols should be defined for these quantities for both plume rise models. Use this sketch to explain in which way the emissions are distributed in the vertical by both models. 2.1 Plume rise model (PRM) Explain what the

basic idea behind this model is, what the input data, and what the output of the model is. 2.2 Plume rise model IS4FIRES Explain what is the basic idea behind this model, what is the input data, and what is the output of the model. 2.3 MISR plume height data set MPHP2 Explain and assess the method by which this data set is produced

3 Integration in GFAS

4. Intercomparison of simulated and observed plume heights

Explain Figure 1

4.1 Comparison of PRM and IS4FIRES results Present and discuss Figure 5 and Figure 6. Present and discuss Table 1 4.2 Comparison with MPHP2

Present and discuss Figure 7 and Figure 8

Present and discuss Table 2 and 3.

In my view Figures 2, 3, and 4 can be removed. If the authors have good reasons to keep them they should be placed here to assess the differences in Figures 7 and 8.

Figures 14 -16 can be removed from the paper. In my view the calculated trends are not significant (prove me if I am wrong). If the authors intend to present the year to year variation of the emission height for both plume rise models they should present Figure 14 and skip figures 15 and 16.

5. Comparison of modelled and measured extinction coefficients for two field campaigns I leave it to the authors if this section is included in the revised version of their paper. If yes they should focus on the extinction coefficient and give a detailed description of how the extinction coefficient is calculated and how they interpret the difference between model results and observations.

6 Summary

Specific comments

Abstract: Explain IS4FIRES. Please check the wording semi-empirical and analytical FRP??PRM. PRM is a numerical not an analytical model. Change 0.1°resolution into 0.1° resolution. Give the name of the new data set of satellite-based plume height observations. Add 'instead of zero plume height or IS4FIRES in the last sentence of the abstract.

Page 2 lines 5-35 These lines are full spelling and grammar errors. Examples are 'Black Carbon' and 'organic carbon', missing or wrong punctuation marks, and wrong style of citations. Extend the checking to the whole paper. Please explain: FLAMBE, GFED, FINN, QFED. Line 22: Modify into 'the recent addition of emission heights'. Line 31: What do you mean by fire smoke releases?

Page 3: Lines 4-6: Please replace these lines and add a sketch as described in the general comments above. Line 6: Maximum injection of what? Line 9: What about static stability and vertical wind shear of the environment? Both quantities are input data for PRM. Line 32: What do you mean by 'coherent climatology'?

Page 4: Line 9: Explain MPHP. Line 11: Your write: This study will be an occasion to revisit their conclusion. Yes indeed, but I did not found this revision in the paper. Line 19: Explain SEAC4RS. Replace 'Forecast' by 'Simulations'.

Page 5: Line 1: What is 'a successful plume' ? Line 17: Replace '20km' by '20 km'. Line 19: What time interval is used for the update of the atmospheric variables taken from ECMWF forecasts? Line 22: Add a multiplication dot to equation 1. Line 23: Explain how the fitting is performed in more detail and give the value of ïĄć used in your study. Line 28: Which injection height is applied in case of smoldering fires?

Page 6: Line 12: CAMS was already explained, so you can use it instead of explaining again.. Line 16: GFAS was already explained. . . Line 20: Explain NASA. Line 20-22 'FRP observations . . . are not used, as they differ from MODIS. Please explain that in more detail. Is it because you just like MODIS more than METEOSAT? Line 23: Replace '°resolution' by '° resolution'. Check 'resp.'

Page 7: Line 3: Replace 'cover' by 'covers'. Line 12-13: Then, the injection profiles. . .
Please explain that in more detail. Line 22: Check '5mn'. Line 24: Replace '°GFAS' by
'° GFAS'. Line 29: Check '5mn'. Line 30: 'maximum' of what?

Page 8: Line 1: What do you mean by density plots? Line 8: Are you talking about
population density? Line 10: Which injection height is used if none is calculated?

Lines 13-30: The usage of injection height by IS4FIRES, mean height of maximum
injection, mean height of injection is totally confusing without having a sketch explaining
the different quantities.

Line 31: Change 'For' into 'for'.

Page 9: Line 1: What do you mean by dispersion? Line 3-10 referring to Figure 4: It is
hard to understand why FRP shows almost no dependence on FRP. Can you explain
this? Have you performed single column sensitivity runs with PRM varying FRP? Line
5: Change 'shows' into 'show'. Line 13: What exactly do you mean by 'injection height
climatologies'?

Page 10: Line 9: I cannot see the day to day variability from figure 6. Line 25: Please
quantify. Is that true in all heights?

Page 11: Line 9: What means 'two times as important as the RMSE'?

Page 12: Line 10: What do you mean by 'smoke fire'? Line 27: How did you quantify
that ECMWF underestimates PBL height?

Page 13: Line 6: What are these six parameters? Line 18: Cams was already ex-
plained. Line 26: You excluded sulphates from 'biomass burning aerosol'. Why?

Page 14: Lines 1-4: You should have explained that earlier. It explains why IS4FIRES
is not used. Section 4.2: You are comparing simulated extinction coefficients with
observations. Which optical properties and which assumptions were used to calculate
the modelled extinction coefficients?

Page 15: Line 9-10: Figure 12 does not show any height of the mixed layer. Line 12: Figure 13 a shows no latitude. Line 15: Where is Figure 13d?

Page 16: Line 30: What follows is rather a Summary than a Conclusion.

Page 17: Line 12-13: How should this optimal combination look like?

Figure 3: What is the difference between injection height and height of maximum injection. What is the' Sofiev height' of injection? It is useless to label the values of a PDF with less dense and dense. Please give numbers, you must have calculated them.

Figure 4: It is useless to label the values of a PDF with less dense and dense. Please give numbers, you must have calculated them. Are you sure that the results of PRM are independent of FRP? This is an input parameter of PRM. In case you are right why is this input parameter needed then?

Figure 6: What is the Sofiev plume height? What is the colour code? What is a density plot? Top figure is not complete.

Figure 7: What means SOFIEV in the legend? Why did you use two reddish colours? This makes it hard to distinguish.

Figure 9: What means PRM mean height? Why do you need a,b,c,d when you indicated these figures by top, bottom etc?

Figure 13: From this Figure it is hard to see that the variable plume height improves the results in comparison with observations.

Figure 16: What is the colour code of the dots? What is the Sofiev injection height?

---

## Referee Comment (RC2) · Anonymous Referee #3 · 29 Aug 2016

General Comments.

The paper addresses the relevant problem of prescribing 4-dimensional (time, lat, long, height) emission fields associated with the biomass burning process, a kind of emission inventory that the atmospheric chemistry modeling community is expecting for a long time. The authors developed a 12 years' climatology of daily biomass burning smoke detrainment layers applying two major approaches referred as the 'plume rise model' (PRM, Paugam et al., 2015b) and the Sofiev's semi-empirical formulation (IS4FIRES, Sofiev et al., 2012). The reviewer has a list of general comments which should be addressed before the final publication in ACP. However, I also have a more philosophical questioning which is directed not only to the authors but also to the handling editor of this manuscript. The manuscript relies on the application of methodologies developed and described by the two works cited above. But, the paper from Paugam et al.

[Figure]

(2015b) was not accepted for the final publication in ACP. That is for me an odd situation. I would recommend a revisit of that manuscript to warrant its publication before the present one be accepted for ACP.

Questions/Comments The text suffers from a significant number of grammatical errors and misspellings words, which prevent them to be listed here. So, the manuscript needs a deep proofreading work.

Pag 3, lines 4-7: The authors should discuss the definition of 'injection height' in the context of the flaming combustion phase. Since during the smoldering phase, a large amount of the smoke can also be produced but it is released just above the surface. Pag 3, line 8. Biomass burning also releases latent heat which also play an important role on the plume buoyancy. Pag 3, line 11: explain what do you meant with 'ambient cooling.'

Pag 4, line 9: Rosário et al. (2013) did not assess various injection height algorithms Page 4, line 9: 'MPHP' must be defined. Page 4, line 30: MPHP2 dataset needs a further reference. Page 5, line 9: The main modifications in versions v1 and v2 of PRM model should be described. Page 5, line 10: State clearly which version exactly the term 'PRM' denotes. Page 5, line 23: State clearly the numerical value of the scaling factor of Eq. 1.

Page 7, lines 15 to 21. The comment does not make sense from the physical point-of-view. The atmospheric stability plays a substantial role on convection either the strongly forced (as above a combustion zone) as well as the weakly forced (e.g., as just above the oceans) situations. If PRM produces a convection plume without the fire forcing, it should produce a deeper plume with the additional buoyancy provided by the fire.

Page 8, lines 5-10. Explain how the smoke emission from the smoldering phase is incorporated in the both methods.

Page 10, lines 26-27: Here a misapplication of the PRM is evident. PRM is 1-d column model and does not account for any lateral mixing associated with the turbulence in the PBL. Any injection layer below the firsts few hundred meters, as shown in figure 7, should be disregard since, in the real world, smoke will be mixed quickly in the PBL. This might be one of the reasons why MISR did not 'see' those shallow plumes. Should be instructive to see RMSE and BIAS without those plumes. In the inventory, the emission associated with the shallow plumes should be just included in the surface level, which will be mixed up nevertheless by the turbulence transport scheme of a 3-d atmospheric model.

---

## Author Response (AR1)

**Answer to the reviews of ACP-2015-1048 « Two global datasets of daily fire emission injection heights since 2003»**

Dear editor, dear reviewers,

Thank you for your review of the manuscript: the numerous corrections and suggestions have led to an improvement in the quality of the paper. We understand the need to restructure the paper and shorten it so as to make its focus clearer. Please note that the title of the paper has been amended. More detailed answers to your comments are detailed below.

Kind regards,
The authors

**General comment by Anonymous Referee #1 :**

> *In the following I will make a detailed suggestion for a revised version and highly encourage the authors to re-submit their important description of the new created input data for the modelling community.*
> *(...)*

The manuscript was restructured along the lines suggested. Figures 2 and 4 have been removed and Figure 3 was reduced in size (4 panels instead of 6), as we felt that the comparison of the output of the PRM and IS4FIRES algorithm against boundary layer height was important to better understand how the two algorithms work.

The description of the C-IFS model has been added to the second section named "Methodology: models and data", with a paragraph on the computation of extinction and optical depth. A sketch has been included to explain in a simple way how the output and input of the two algorithms.

Figures 14-16 and the corresponding section have been removed from the manuscript.

**Specific comments :**

> *Abstract: Explain IS4FIRES. Please check the wording semi-empirical and analytical FRP??PRM. PRM is a numerical not an analytical model. Change 0.1*
> *◦ resolution into 0.1◦resolution. Give the name of the new data set of satellite-based plume height observations. Add 'instead of zero plume height or IS4FIRES in the last sentence of the abstract.*

The abstract has been modified and corrected along these lines. IS4FIRES is not really an acronym; the full name of the model was given.

Page 2 lines 5-35 These lines are full spelling and grammar errors. Examples are
'Black Carbon' and 'organic carbon', missing or wrong punctuation marks, and wrong style of
citations. Extend the checking to the whole paper. Please explain: FLAMBE, GFED, FINN,
QFED. Line 22: Modify into 'the recent addition of emission heights'. Line 31: What do you mean
by fire smoke releases

Page 3: Lines 4-6: Please replace these lines and add a sketch as described in the
general comments above. Line 6: Maximum injection of what? Line 9: What about
static stability and vertical wind shear of the environment? Both quantities are input data for
PRM. Line 32: What do you mean by 'coherent climatology'

The sketch has been added and the corresponding lines removed. Static stability and vertical wind shear have been added. "Coherent climatology" was to emphasize that emissions and injection heights were gridded and assimilated in the same way. This paragraph has been entirely rewrittent, and the term "dataset" was preferred to "climatology".

Page 4: Line 9: Explain MPHP. Line 11: Your write: This study will be an occasion to
revisit their conclusion. Yes indeed, but I did not found this revision in the paper. Line 19: Explain
SEAC4RS. Replace 'Forecast' by 'Simulations'

Corrected, thank you. The sentence *"This study will be an occasion to revisit their conclusion"* has been removed as it is not really the main focus of this work. A mention to the conclusions from Val Martin et al (2012) and Strada et al (2013) was nonetheless added in the discussion section.

Page 5: Line 1: What is 'a successful plume'? Line 17: Replace '20km' by '20 km'.
Line 19: What time interval is used for the update of the atmospheric variables taken from
ECMWF forecasts? Line 22: Add a multiplication dot to equation 1. Line 23: Explain how the
fitting is performed in more detail and give the value of ï¿A
used in your study. Line 28: Which injection height is applied in case of smoldering fires

Line 1, 17 and 22: corrected, thank you.
Line 19: the interval is 3 hours, this information has been added.
Line 23: the fitting is done using two algorithms sequentially: simulated annealing and Markov Chain Monte Carlo. This was not added into the text but to an Annex A which is a summary of Paugam et al 2015b and described the changes of PRMv2 as compared to v0.
Line 28: in case of smoldering fires, it is advised to prescribe biomass burning emissions at the surface. This has been added in this section and also in the section that describes the integration of the PRM into GFAS.

Page 6: Line 12: CAMS was already explained, so you can use it instead of explaining again..
Line 16: GFAS was already explained
...Line 20: Explain NASA. Line 20-22 'FRP observations are not used, as they differ from MODIS.
Please explain that in more detail. Is it because you just like MODIS more than METEOSAT? Line
23: Replace '∘resolution' by '∘resolution'. Check 'resp.'

Line 12, 16 and 23: corrected, thank you.

Line 20-22: one more sentence has been added to explain why geostationary data is not (yet) used in GFAS.
* * *
*Page 7: Line 3: Replace 'cover' by 'covers'. Line 12-13: Then, the injection profiles ...Please explain that in more detail. Line 22: Check '5mn'. Line 24: Replace '∘GFAS' by '∘GFAS'. Line 29: Check '5mn'. Line 30: 'maximum' of what?*
* * *
Line 3, 22, 24, 30: corrected thank you
Line 12-13: this sentence was rewritten and completed to try to make it clearer.
* * *
*Page 8: Line 1: What do you mean by density plots? Line 8: Are you talking about population density? Line 10: Which injection height is used if none is calculated?*
* * *
Line 1 and 8: density plots are an alternative to scatterplots when there are too many points to plot. The underlying idea is to plot density of points rather than the points themselves. This allows to see better the correlations and shapes for large samples.
Line 10: In C-IFS, when GFAS provides no injection height then biomass burning emissions are prescribed at the surface. A sentence was added there to carry this message.
* * *
*Lines 13-30: The usage of injection height by IS4FIRES, mean height of maximum injection, mean height of injection is totally confusing without having a sketch explaining the different quantities.*
* * *
A sketch has been added to explain the mean height of maximum injection from the PRM and injection height from IS4FIRES.
* * *
*Line 31: Change 'For' into 'for*
* * *
Done, thank you.
* * *
*Page 9: Line 1: What do you mean by dispersion? Line 3-10 referring to Figure 4: It is hard to understand why FRP shows almost no dependence on FRP. Can you explain this? Have you performed single column sensitivity runs with PRM varying FRP? Line 5: Change 'shows' into 'show'. Line 13: What exactly do you mean by 'injection height climatologies'?*
* * *
This part has been removed from the manuscript.
The lack of dependence on FRP was indeed a surprise, as other studies found that FRP is having a strong impact on the output of the PRM (ValMartin et al 2012). We have not performed single column sensitivity runs with varying FRP as an input. In our implementation of the PRM, it

the output was much more dependent on the ambient condition (atmospheric stability especially) than on FRP. Possibly with fixed ambient conditions the impact of the FRP input might be more important, but this has not been explored as we felt it was outside the scope of this paper (which is to integrate existing algorithms into GFAS, evaluate the resulting datasets and test the output in C-IFS).

*Page 10: Line 9: I cannot see the day to day variability from figure 6. Line 25: Please quantify. Is that true in all heights?*
*Page 11: Line 9: What means 'two times as important as the RMSE'?*
*Page 12: Line 10: What do you mean by 'smoke fire'? Line 27: How did you quantify that ECMWF underestimates PBL height?*

Page 10, line 9 and Page 11, line 9, corrected, thank you.
Page 12 line 10, it was a typo, corrected.
Page 12 line 27, this sentence was modified as the reasoning goes the other way round: since the PRM show a strong dependence on PBL height (see Figure 3), the fact that there is a significant subset of the output with very important underestimation of the plume height as compared to observation, it is a good sign that in some cases the ECMWF PBL diagnostic could give lower values as compared to what observations would provide. These could correspond to night-time values, where it is a known problem of most turbulent schemes that the lower boundary layer is too stable (and thus the PBL height too low compared to observations). This is only speculation, and no observations exist to give ground to this possible explanation.

*Page 13: Line 6: What are these six parameters? Line 18: Cams was already explained. Line 26: You excluded sulphates from 'biomass burning aerosol'. Why?*

The six parameters are detailed in a new Annex A.
Sulphates was excluded from biomass burning aerosol because there are many other sources for this particular species. The definition "biomass burning aerosol" is used globally at ECMWF.

*Page 14: Lines 1-4: You should have explained that earlier. It explains why IS4FIRES is not used. Section 4.2: You are comparing simulated extinction coefficients with observations. Which optical properties and which assumptions were used to calculate the modelled extinction coefficients?*

Section 4.2 was moved to Section 2 and a paragraph was added about how the optical properties of aerosols are computed in the C-IFS model.

*Page 15: Line 9-10: Figure 12 does not show any height of the mixed layer. Line 12: Figure 13 a shows no latitude. Line 15: Where is Figure 13d? Page 16: Line 30: What follows is rather a Summary than a Conclusion. Page 17: Line 12-13: How should this optimal combination look like?*

Page 15 line 9-10, corrected thank you, this referred to another version of this plot.

Page 15 line 12: this was inferred from Figure 12; the sentence was corrected.

Page 17: this optimal dataset would have a larger variability than the IS4FIRES dataset, and a smaller error/bias than the PRM dataset. This was added to the conclusion.

*Figure 3: What is the difference between injection height and height of maximum injection. What is the' Sofiev height' of injection? It is useless to label the values of a PDFwith less dense and dense. Please give numbers, you must have calculated them.*
*Figure 4: It is useless to label the values of a PDF with less dense and dense. Please give numbers, you must have calculated them. Are you sure that the results of PRM are independent of FRP? This is an input parameter of PRM. In case you are right whyis this input parameter needed then?*

Figure 3: injection height is the output of the IS4FIRES algorithm (and corresponds more to the top of the plume); mean height of maximum injection is an output of the PRM algorithm (and corresponds to the height at which injection is maximal). Two panels has been removed from this figure, and the labels have been removed. Unfortunately there are no numbers to be put on the labels since the computing algorithm only provides density values from 0 to 1.

Figure 4 has been removed from the new version of the manuscript. As discussed above, it is possible that with similar atmospheric profiles, the impact of the FRP input would be larger on the output of the PRM.

*Figure 6: What is the Sofiev plume height? What is the colour code? What is a density plot? Top figure is not complete.*
*Figure 7: What means SOFIEV in the legend? Why did you use two reddish colours? This makes it hard to distinguish.*

Figure 6: the labels has been corrected and the top figure moved slightly

Figure 7: SOFIEV replaced by IS4FIRES in the legend.

*Figure 9: What means PRM mean height? Why do you need a,b,c,d when you indicated these figures by top, bottom etc?*
*Figure 13: From this Figure it is hard to see that the variable plume height improves the results in comparison with observations.*
*Figure 16: What is the colour code of the dots? What is the Sofiev injection height?*

Figure 9: the label has been modified and the letters removed

Figure 13: the improvement is marginal, this has been added into the text. As mentioned also, there are numerous sources of errors/uncertainties in these simulations of aerosol extinction

besides the injection height.

Figure 16 has been removed from the manuscript.

**General comment by Anonymous Referee #3 :**

*However, I also have a more philosophical questioning which is directed not only to the authors but also to the handling editor of this manuscript. The manuscript relies on the application of methodologies developed and described by the two works cited above. But, the paper from Paugam et al.(2015b) was not accepted for the final publication in ACP. That is for me an odd situation. I would recommend a revisit of that manuscript to warrant its publication before the present one be accepted for ACP.*

The paper of Paugam et al. (2015b) describes a new version of the PRM with two additions: a new entrainment scheme and a mass conservation equation. However the PRM itself is already described in the two papers of Freitas et al. (2007, 2010), and has been used in other studies (ValMartin, et al. 2012, Strada et al. 2013). Pending a revisit of the Paugam et al (2015b) paper, we have reverted to cite the earlier works of Freitas et al. An annex was also added to summarize the changes brought by Paugam et al (2015b).

The PRM model is not fundamentally different in our implementation as compared to the implementation of ValMartin et al (2012) for example.

*Pag 3, lines 4-7: The authors should discuss the definition of 'injection height' in the context of the flaming combustion phase. Since during the smoldering phase, a large amount of the smoke can also be produced but it is released just above the surface.*
*Pag 3, line 8. Biomass burning also releases latent heat which also play an important role on the plume buoyancy. Pag 3, line 11: explain what do you meant with 'ambient cooling.'*

Line 4-7: The injection heights are indeed, in this work, only meant for flaming fires. For smoldering fires, it is advised to prescribe emissions at the surface. This mention was lacking in the manuscript and was added in section 2 and section3

Line 8: It was already mentioned but not clear enough: a sentence was added about latent heat release

Line 11:This part was badly worded and has been modified, thank you.

*Pag 4, line 9: Rosário et al. (2013) did not assess various injection height algorithms*
*Page 4, line 9: 'MPHP' must be defined. Page 4, line 30: MPHP2 dataset needs a*
*further reference. Page 5, line 9: The main modifications in versions v1 and v2 of PRM model should be described. Page 5, line 10: State clearly which version exactly the term 'PRM' denotes.*
*Page 5, line 23: State clearly the numerical value of the scaling factor of Eq. 1.*

Page 4 line 9: corrected, thank you.

Page 4 line 30: unfortunately there is not yet a reference for the MPHP2 dataset

page 5 line 9: the modifications of PRMv2 as compared to PRMv0 are described in a new Annex.

Page 5 line 10: In this work PRMv2 is used, it has been emphasized again in section 2.

Page 5 line 23: the value of this scaling factor (and of the other 5 parameters used in the PRM) is shown in the new Annex A

*Page 7, lines 15 to 21. The comment does not make sense from the physical point-of-view. The atmospheric stability plays a substantial role on convection either the strongly forced (as above a combustion zone) as well as the weakly forced (e.g., as just above the oceans) situations. If PRM produces a convection plume without the fire forcing, it should produce a deeper plume with the additional buoyancy provided by the fire*

In a number of cases (around 10% of the total), the PRM produces the same plume with and without fire forcing at its base. These cases that are removed from the final injection height database.

*Page 8, lines 5-10. Explain how the smoke emission from the smoldering phase is incorporated in the both methods.*

The smoldering phase is not incorporated: a test on fire temperature and fire area tries to select only fires in the flaming phase. For smoldering fires, which emit mostly at the surface, there is no output from the PRM or IS4FIRES. Two mentions (in section 2 and 3) have been added on this subject.

*Page 10, lines 26-27: Here a misapplication of the PRM is evident. PRM is 1-d column model and does not account for any lateral mixing associated with the turbulence in the PBL. Any injection layer below the firsts few hundred meters, as shown in figure 7, should be disregard since, in the real world, smoke will be mixed quickly in the PBL. This might be one of the reasons why MISR did not 'see' those shallow plumes. Should be instructive to see RMSE and BIAS without those plumes. In the inventory, the emission associated with the shallow plumes should be just included in the surface level, which will be mixed up nevertheless by the turbulence transport scheme of a 3-d atmospheric model.*

The plots in Figures 6 and 7 are all done for fires which has a MPHP2 observation of fair or good quality: this means that the low values given by the PRM are associated with much higher retrieved values from MPHP2 (the legend of the figure was updated as this fact was not made clear).

We agree that these heights should not be used, which is the case in our implementation of the use of biomass burning injection heights in C-IFS. In practice the most important information provided by the injection heights is whether they are above or under the PBL height.

The scores without these low values have been computed: they are not too different (global RMSE decreases from 955m to 930m; bias increases from 239m to 355m)